# Prognosis and Interplay of Cognitive Impairment and Sarcopenia in Older Adults Discharged from Acute Care Hospitals

**DOI:** 10.3390/jcm8101693

**Published:** 2019-10-15

**Authors:** Elisa Zengarini, Robertina Giacconi, Lucia Mancinelli, Giovanni Renato Riccardi, Daniele Castellani, Davide Liborio Vetrano, Graziano Onder, Stefano Volpato, Carmelinda Ruggiero, Paolo Fabbietti, Antonio Cherubini, Francesco Guarasci, Andrea Corsonello, Fabrizia Lattanzio

**Affiliations:** 1Geriatric Medicine and Geriatric Emergency Care, Italian National Research Center on Aging (IRCCS INRCA), 60127 Ancona, Italy; e.zengarini@inrca.it (E.Z.); a.cherubini@inrca.it (A.C.); 2Translational Research Center of Nutrition and Ageing, IRCCS INRCA, 60127 Ancona, Italy; r.giacconi@inrca.it; 3Department of Cardiology, IRCCS INRCA, 60127 Ancona, Italy; l.mancinelli@inrca.it; 4Posture and Movement Analysis Laboratory, IRCCS INRCA, 60127 Ancona, Italy; g.riccardi@inrca.it; 5Department of Urology, IRCCS INRCA, 60127 Ancona, Italy; 6Aging Research Center, Department of Neurobiology, Care Sciences and Society, Karolinska Institutet and Stockholm University, 11330 Stockholm, Sweden; davide.vetrano@ki.se; 7Department of Cardiovascular, Metabolic and Aging Diseases, Istituto Superiore di Sanità, 00161 Rome, Italy; graziano.onder@unicatt.it; 8Department of Medical Sciences, Section of Internal and Cardiorespiratory Medicine, University of Ferrara, 44121 Ferrara, Italy; 9Section of Gerontology and Geriatrics, Department of Medicine, University of Perugia, 06156 Perugia, Italy; carmelinda.ruggiero@unipg.it; 10Unit of Geriatric Pharmacoepidemiology, IRCCS INRCA, 60127 Ancona, Italy; p.fabbietti@inrca.it (P.F.); f.guarasci@inrca.it (F.G.); a.corsonello@inrca.it (A.C.); 11Italian National Research Center on Aging, Scientific Direction, IRCCS INRCA, 60127 Ancona, Italy; f.lattanzio@inrca.it

**Keywords:** sarcopenia, cognitive impairment, mortality, hospital, older patients

## Abstract

Sarcopenia and cognitive impairment are associated with an increased risk of negative outcomes, but their prognostic interplay has not been investigated so far. We aimed to investigate the prognostic interaction of sarcopenia and cognitive impairment concerning 12-month mortality among older patients discharged from acute care wards in Italy. Our series consisted of 624 patients (age = 80.1 ± 7.0 years, 56.1% women) enrolled in a prospective observational study. Sarcopenia was defined following the European Working Group on Sarcopenia in Older People (EWGSOP) criteria. Cognitive impairment was defined as age- and education-adjusted Mini-Mental State Examination (MMSE) score < 24 or recorded diagnosis of dementia. The study outcome was all-cause mortality during 12-month follow-up. The combination of sarcopenia and cognitive ability was tested against participants with intact cognitive ability and without sarcopenia. Overall, 159 patients (25.5%) were identified as having sarcopenia, and 323 (51.8%) were cognitively impaired. During the follow-up, 79 patients (12.7%) died. After adjusting for potential confounders, the combination of sarcopenia and cognitive impairment has been found associated with increased mortality (HR = 2.12, 95% CI = 1.05–4.13). Such association was also confirmed after excluding patients with dementia (HR = 2.13, 95% CI = 1.06–4.17), underweight (HR = 2.18, 95% CI = 1.03–3.91), high comorbidity burden (HR = 2.63, 95% CI = 1.09–6.32), and severe disability (HR = 2.88, 95% CI = 1.10–5.73). The co-occurrence of sarcopenia and cognitive impairment may predict 1-year mortality in older patients discharged from acute care hospitals.

## 1. Introduction

Sarcopenia is a progressive and generalized skeletal muscle disorder (i.e., muscle failure) rooted in adverse muscle changes occurring across a lifetime, with low muscle strength as principal determinant [1,2,3]. Although sarcopenia is also present in youngers, it is a highly prevalent geriatric syndrome: 5% to 13% among people aged 60–70 years, and 11% to 50% among those aged 80 years or older [2,4,5]. Sarcopenia is known to herald adverse outcomes in different care settings, including hospitals. Indeed, sarcopenia was found to increase the risk of disability, falls, hospitalization, and mortality in several different populations [6,7,8,9,10].

Sarcopenia can be categorized into primary and secondary forms. Primary sarcopenia is age-related and no evident factors are present. When other causal factors are present, sarcopenia is considered secondary. Systemic diseases (malignant or inflammatory), physical inactivity (sedentary lifestyle, disability, or disease-related immobility), neurological disorders, and malnutrition can contribute to sarcopenia development [11]. Among the molecular pathways contributing to sarcopenia, mitochondrial dysfunction, oxidative stress, chronic inflammation, and hormonal changes are the main mechanisms involved. The maintenance of mitochondrial function is especially relevant to myocyte viability because of their high reliance on oxidative metabolism for energy production. Downregulation of mitochondrial biogenesis, oxygen consumption, and ATP production have been observed in physically inactive elderly [12]. Moreover, mitochondria are the primary source of oxidative stress, which may contribute to enhance the release of pro-inflammatory cytokines that, inducing muscle catabolism, may promote sarcopenia development [13].

Sarcopenia was found associated with cognitive impairment in both cross-sectional [14] and longitudinal studies [15], although some controversies are reported [16]. Skeletal muscle is known to secrete neurotrophic factors able to affect brain function and motor units in skeletal muscle [17]. On the other hand, the central nervous system plays a crucial role in muscle integrity maintenance in older people [18]. Several age-related changes in the nervous system, including downregulated dopaminergic neurotransmission and inherent decline of supraspinal drive, inflammation, and remodeling of neuromuscular junction, may affect motor performance, muscle strength, and muscle mass [19]. Sarcopenia and cognitive impairment share several common mechanisms. Inflammation may accelerate muscle protein breakdown and muscle wasting through several pathophysiologic pathways, including insulin resistance, adipose tissue accumulation, and oxidative stress [20]. On the other hand, the accumulation of β-amyloid in the brain is known to stimulate an inflammatory response with increased TNF-α production and consequent blunted protein synthesis, impaired insulin signaling, synapse deregulation, and cognitive impairment [21]. Estrogens and androgens are known to influence the growth and maintenance of muscle, and their age-related decline may contribute to loss of muscle mass and functional integrity [22]. Sexual neurosteroids also play a crucial role in the maintenance of synaptic plasticity [23], and decline in sex hormones has been found to predict cognitive impairment [24,25]. Selected nutritional deficits may affect both muscle and cognitive function. Vitamin D impacts muscle mass and functionality [26], and growing evidence recognizes that vitamin D is linked to brain development, functions, and diseases [27]. Additionally, a recent meta-analysis including 26 observational studies clearly indicates that low vitamin D is associated with poorer cognition [28].

Despite this bulk of evidence, the prognostic significance of the interplay of cognitive impairment and sarcopenia has not been investigated until now. Therefore, we aimed to investigate whether the association between sarcopenia and mortality may change as a function of the presence or absence of cognitive impairment in a cohort of older patients admitted to acute care wards in Italy.

## 2. Materials and Methods

### 2.1. Study Design and Participants

We analyzed data from the CRiteria to Assess Appropriate Medication Use among Elderly Complex Patients (CRIME) project, which was a prospective observational study carried out in the geriatric and internal medicine acute care wards of seven Italian hospitals. The methodology of the CRIME project has been previously described in detail [29]. Briefly, all patients consecutively admitted to seven participating wards between June 2010 and May 2011 were asked to take part in the study. Exclusion criteria were age less than 65 years and unwillingness to participate in the study. After obtaining written informed consent, all participants were assessed within the first 24 h from hospital admission and followed until discharge. Information was collected on demographics, socioeconomic, and clinical characteristics, with comprehensive data collection on pharmacological therapy, geriatric syndromes, and comprehensive geriatric assessment. After discharge, patients were reassessed at 3, 6, and 12 months. All subjects gave their informed consent for inclusion before they participated in the study. The study was conducted in accordance with the Declaration of Helsinki, and the protocol was approved by the Ethics Committee of the Catholic University of Rome (Project identification code: P/582/CE/2009).

### 2.2. Outcome

The outcome of the present study was 1-year mortality. Data on living status during follow-up were obtained by interviewing the patients and/or their formal and/or informal caregivers. For patients who died during the follow-up period, the date and place of death were retrieved by certificates of death exhibited by relatives or caregivers.

### 2.3. Exposure Variables

The European Working Group on Sarcopenia in Older People (EWGSOP) criteria were used to diagnose sarcopenia [1]. According to the EWGSOP recommendations, the presence of low muscle mass plus either low muscle strength or low physical performance was required for the diagnoses of sarcopenia. Therefore, to assess sarcopenia at discharge, the following parameters at hospital discharge were considered:

(a) *Handgrip measurement.* Muscle strength was tested through the handgrip strength, which was measured using a dynamometer (North Coast Hydraulic Hand Dynamometer, North Coast Medical Inc, Morgan Hill, CA) with the patient seated with the wrist in a neutral position, and the elbow flexed 90°. For patients unable to sit, grip strength was assessed lying at 30° in bed with elbows supported [30]. Two trials for each hand were performed and the best result from the strongest hand chosen. Using the cut-off indicated in the EWGSOP definition, low muscle strength was classified as handgrip less than 30 kg in men and less than 20 kg in women.

(b) *Muscle mass assessment*. Muscle mass was measured by bioelectrical impedance analysis (BIA), a well-studied and validated technique to measure body composition and predict muscle mass. The BIA resistance was obtained using a Quantum/S Bioelectrical Body Composition Analyzer (Akern Srl, Florence, Italy) with an operating frequency of 50 kHz at 800 mA. Whole-body BIA measurements were taken between the right wrist and ankle with the subject in a supine position. BIA was not performed in patients with peripheral edema and among those with pacemaker or implantable cardioverter-defibrillator. Muscle mass was calculated using the BIA equation by Janssen and colleagues [31]. The skeletal muscle index (SMI (kg/m^2^)) was obtained dividing absolute muscle mass for squared height. According to the EWGSOP criteria, low muscle mass was classified as an SMI less than 8.87 kg/m^2^ in men and less than 6.42 kg/m^2^ in women. SMI was not calculated in patients unable to stand, because height could not be measured.

(c) *Gait speed measurement.* Walking speed was evaluated measuring the participant’s usual gait speed over a 4-meter course. As recommended in the EWGSOP consensus, low physical performance was defined by a gait speed less than or equal to 0.8 meters per second (m/s).

Analyses were also repeated by using updated version 2 of EWGSOP criteria [3], including (a) handgrip strength less than 27 kg in men and less than 16 kg in women, and (b) SMI less than 7 kg/m^2^ in men and less than 5.5 kg/m^2^ in women.

Cognitively impaired patients were identified by age- and education-adjusted Mini-Mental State Examination score less than 24 [32] or recorded diagnosis of dementia. The assessment of cognitive status was carried out in stable condition (usually the day before discharge) by a study physician.

The primary analytical variable was created by grouping patients as follows: no sarcopenia and no cognitive impairment, cognitive impairment and no sarcopenia, sarcopenia and no cognitive impairment, and sarcopenia and cognitive impairment.

### 2.4. Covariates

Covariates included sociodemographic variables (i.e., age and gender) and other information derived from a comprehensive geriatric assessment. Functional status was defined by the number of Basic Activities of Daily Living (BADL) lost at discharge [33]. Patients with a Geriatric Depression Scale score > 5 were considered depressed [34]. Underweight status (defined by a body mass index (BMI) less than 20 kg/m^2^), number of diagnoses, number of medications at discharge, and length of stay were also included in the analyses.

The following diseases and geriatric syndromes were also considered in the analysis: hypertension, coronary artery disease, atrial fibrillation, peripheral artery disease, heart failure, cerebrovascular disease, dementia, Parkinson’s disease, diabetes, chronic obstructive pulmonary disease, chronic kidney disease (defined by a glomerular filtration rate (GFR) estimated by Berlin Initiative Study (BIS) [35] equation less than 60 mL/min/1.73m^2^), malignancies, history of falls, urinary incontinence, and delirium during stay.

### 2.5. Sample Selection

Overall, 1123 patients were enrolled in the study. Patients with incomplete baseline data (N = 3) and those who died during hospitalization (N = 39) were excluded from the present analysis. Patients with missing data on sarcopenia (N = 334) and cognitive assessment (N = 25) were also excluded, as were patients with incomplete follow-up data (N = 98), leaving a final sample of 624 participants to be included in the analysis. Patients excluded from the study because of missing data were older (age 82.8 ± 7.5, *p* < 0.001) and had a higher prevalence of dependency in at least 1 BADL (52.3%, *p* < 0.001) and urinary incontinence (43.3%, *p* < 0.001) compared to included ones.

### 2.6. Analytic Approach

First, we compared patients’ baseline demographic and clinical characteristics across the four groups based on the presence/absence of sarcopenia and cognitive impairment. One-way ANOVA was used for continuous variables and chi-square test for categorical ones.

Therefore, we built three different Cox proportional hazard models to test the association between exposure variable and mortality: unadjusted; adjusted for age and gender; and adjusted for age, gender, dependency in at least 1 BADL, depression, BMI < 20 kg/m^2^, history of falls, urinary incontinence, delirium during stay, number of diagnoses, number of medications, and length of stay. This latter model was also repeated after including the above-listed selected diagnoses, instead of the number of comorbidities. The interaction term sarcopenia*cognitive impairment was also included in the model. Sensitivity analyses were also carried out by running the above-described fully adjusted model, after excluding patients with recorded diagnosis of dementia, BMI < 20 kg/m^2^, more than 5 diagnoses, or severe disability.

All statistical analyses were performed with SPSS version 23 (SPSS Inc., Chicago, IL, USA).

## 3. Results

The mean age of the study participants was 80.1 ± 7.0 years, and 350 participants (56.1%) were women. According to the EWGSOP criteria, 159 subjects (25.5%) were identified as having sarcopenia. Three-hundred and twenty-three patients (51.8%) presented cognitive impairment, and 82 of them had recorded diagnosis of dementia. The prevalence of cognitive impairment was similar among sarcopenic (54.7%) and non-sarcopenic patients (50.7%, *p* = 0.373). Demographic and clinical characteristics of patients grouped according to the presence of sarcopenia and cognitive impairment are shown in Table 1. Compared to patients with no cognitive impairment and no sarcopenia, those with cognitive impairment and no sarcopenia were older and more frequently females. Individuals with BADL dependency, depression, history of falls, urinary incontinence, underweight, and cerebrovascular disease were also more prevalent among non-sarcopenic but cognitively impaired patients. Patients with sarcopenia and no cognitive impairment were characterized by older age, lower prevalence of female gender and cerebrovascular disease, and higher prevalence of depression, underweight, and malignancies compared to those without cognitive impairment and sarcopenia. Finally, patients affected by combined sarcopenia and cognitive impairment were older and more often female and had a higher prevalence of BADL dependency, history of falls, urinary incontinence, underweight, heart failure, and cerebrovascular disease compared to those with no sarcopenia and no cognitive impairment (Table 1).

During the follow-up period, 79 patients (12.7%) died. Kaplan–Meier curves showed that participants with cognitive impairment alone or sarcopenia alone had similar shorter survival during 1-year follow-up period, while patients with both sarcopenia and cognitive impairment had the lowest survival rate (Log rank = 16.812, *p* < 0.001) (Figure 1).

After adjusting for potential confounders, only the presence of both sarcopenia and cognitive impairment was significantly associated with mortality (Table 2). Age (HR = 1.06, 95% CI = 1.02–1.13) and dependency in at least 1 BADL (HR = 2.60, 95% CI = 1.53–4.44) also qualified as predictors of the outcome. When further adjusting for individual diagnoses included in the analysis instead of number of diagnoses, the association of cognitive impairment and/or sarcopenia with mortality was substantially unchanged (cognitive impairment without sarcopenia: HR = 1.50, 95% CI = 0.80–2.84; sarcopenia without cognitive impairment: HR = 1.79, 95% CI = 0.88–3.84; combined sarcopenia and cognitive impairment: HR = 2.33, 95% CI = 1.12–4.81). Other significant predictors in this latter model were age (HR = 1.05, 95% CI = 1.01–1.09), dependency in at least 1 BADL (HR = 2.71, 95% CI = 1.58–4.66), and malignancies (HR = 4.47, 95% CI = 2.71–7.38). The interaction term sarcopenia*cognitive impairment on the risk of mortality was statistically significant (HR = 2.10, 95% CI = 1.26–3.47, *p* = 0.004).

We also re-run multivariable models after changing the reference category. When considering cognitive impairment and no sarcopenia as reference, the association between combined sarcopenia and cognitive impairment was still near significant (HR = 1.63, 95% CI = 0.95–2.84), while such an association was no longer present when considering sarcopenia and no cognitive impairment as reference category (HR = 1.46, 95% CI = 0.70–3.05).

The association of co-occurrence of sarcopenia and cognitive impairment with mortality was also confirmed in sensitivity analyses carried out after excluding patients with a recorded diagnosis of dementia, underweight, high comorbidity burden, and severe disability. Among patients without severe disability, sarcopenia without cognitive impairment was also significantly associated with mortality, and a non-significant trend for association was also observed for cognitive impairment without sarcopenia (Table 2).

When using EWGSOP2 criteria, 98 subjects (15.7%) were identified as having sarcopenia. Of them, 61 also had cognitive impairment. The association of combined sarcopenia and cognitive impairment with mortality was confirmed in the fully adjusted Cox regression model (cognitive impairment and no sarcopenia HR = 1.43, 95% CI = 0.80–2.54; sarcopenia and no cognitive impairment HR = 2.04, 95% CI = 0.86–5.54; sarcopenia and cognitive impairment HR = 2.25, 95% CI = 1.09–4.56). Though with a lower strength, the interaction term sarcopenia*cognitive impairment was statistically significant even in this analysis (HR = 2.08, 95% CI = 1.08–3.70, *p* = 0.013).

## 4. Discussion

The present study adds to the current knowledge that the co-occurrence of sarcopenia and cognitive impairment may predict mortality in an unselected population of older patients discharged from acute care hospitals. Several studies have shown that sarcopenia is associated with increased mortality in older hospitalized patients [7,36], nursing home residents [37], and in community-dwelling individuals [38]. At the same time, several population-based longitudinal studies showed that cognitive impairment is a significant independent predictor of mortality [39,40]. Moreover, a recent prospective study showed that frailty and cognitive impairment were associated with an increased mortality rate, with significant interaction between frailty and cognitive decline [41]. However, to the best of our knowledge, the present study is the first providing evidence that sarcopenia, cognitive impairment, or their combination, may impact differently on survival.

Interestingly, the recently described motoric-cognitive risk (MCR) syndrome, defined as the simultaneous presence of gait disturbance and mild cognitive impairment [42], was found to predict adverse outcomes among older adults, including disability [43] and death [44]. Further confirming this view, Marengoni et al. recently showed that combined slow gait speed and impaired memory might predict mortality in community-dwelling older people [45]. Finally, disability was a relevant confounder in our study. Given the well-known impact of both sarcopenia and cognitive impairment on functional status, it can be argued that disability may represent another important mechanism in their prognostic interplay.

Whatever is the mechanism underlying the increased risk of death observed among older patients with combined sarcopenia and cognitive impairment, our study results underline the importance of identifying and treating hospitalized patients affected by cognitive impairment and sarcopenia. The routine assessment of sarcopenia should be implemented in clinical practice, as advocated repeatedly by several scientific societies. It is noteworthy that sarcopenia is now formally recognized as a muscle disease with an ICD-10-MC Diagnosis Code that can stimulate its detection. Regarding the effective treatments, there is substantial evidence suggesting that exercise interventions with or without nutritional supplementation may improve physical performance, and multidisciplinary interventions including exercise may improve muscle strength in older people with frailty and sarcopenia [46,47]. Furthermore, rising evidence indicates that combined intervention of physical activity and cognitive training may have positive effects on mental and physical functions in patients with Mild Cognitive Impairment [48,49]. Nevertheless, the effectiveness of such interventions in patients with combined sarcopenia and cognitive impairment is mostly unknown and warrants future investigations. It is worth noting that our findings also showed that combined sarcopenia and cognitive impairment might increase mortality when compared to cognitive impairment alone, but not sarcopenia alone, which may have relevant clinical implications. Indeed, if we consider that sarcopenia may worsen the prognostic impact of cognitive impairment, preventing loss of muscle mass and function would represent an essential strategy in the management of cognitively impaired patients.

Finally, after excluding patients with severe disability, sarcopenia without cognitive impairment and, to a lesser extent, cognitive impairment without sarcopenia emerged as mortality predictors in our study, as if the prognostic role of individual risk factors was more significant among patients who were less disabled at the time of discharge. Thus, our findings did not reduce the relevance of sarcopenia and cognitive impairment as individual risk factors, instead contributed to describing a condition of particular vulnerability being targeted by specific interventions.

The present study has several limitations. First, the primary aim of the CRIME project did not include the investigation of the role of sarcopenia and cognitive impairment on clinically relevant outcomes. Thus, the definition of sarcopenia and cognitive impairment were obtained a posteriori and adapted to the purpose of the present study. Second, patients’ acute conditions related to hospitalization may have contributed to an overestimation of the diagnosis of sarcopenia and cognitive impairment in the present sample. Indeed, transient symptoms including weakness and mental confusion may influence the performance on physical and cognitive tests, and changes in hydration status may affect body composition. Additionally, variation of acute illness and discharge destination might influence prognosis of studied patients. Third, our findings may be influenced by residual confounding from other not explored variables, such as the overall quality of post-discharge care, as well as the role of formal and informal caregivers in the assistance of older patients about nutritional status, social involvement, and physical activity. Additionally, patients with combined sarcopenia and cognitive impairment were older and had a higher prevalence of selected risk factors compared to those without cognitive impairment or sarcopenia. Thus, even if our results were also confirmed after the exclusion of individuals with recorded diagnoses of dementia, multimorbidity, or severe disability, confounding by indication may represent a significant limitation in our study. Patients excluded because of missing data were older and had greater prevalence of dependency and urinary incontinence compared to included ones. Thus, our findings may not apply to the general population of older hospitalized patients. The small sample size may also represent a relevant limitation to be acknowledged, especially in sensitivity analyses. Additionally, it prevented us from accounting for sarcopenia severity when considering EWGSOP2 criteria. Finally, data about the cause of mortality were not collected, which may limit the understanding of the different predictive values of sarcopenia and/or cognitive impairment. Nevertheless, strengths of our study are the inclusion of an unselected population of older hospitalized patients, the longitudinal design, the systematic use of comprehensive geriatric assessment, which allowed to consider several and relevant confounders, and the rigorous analytical method.

## 5. Conclusions

The co-occurrence of sarcopenia and cognitive impairment may predict 1-year mortality in older patients discharged from acute care hospitals. This finding confirms the close interaction between the physical and cognitive domains as well as the need for a comprehensive approach to identify hospitalized patients at high risk of adverse outcomes. The pathophysiology shared between cognitive impairment and sarcopenia, the impact on outcomes other than mortality, as well as effective preventive interventions warrant further investigations.

## Figures and Tables

**Figure 1 jcm-08-01693-f001:**
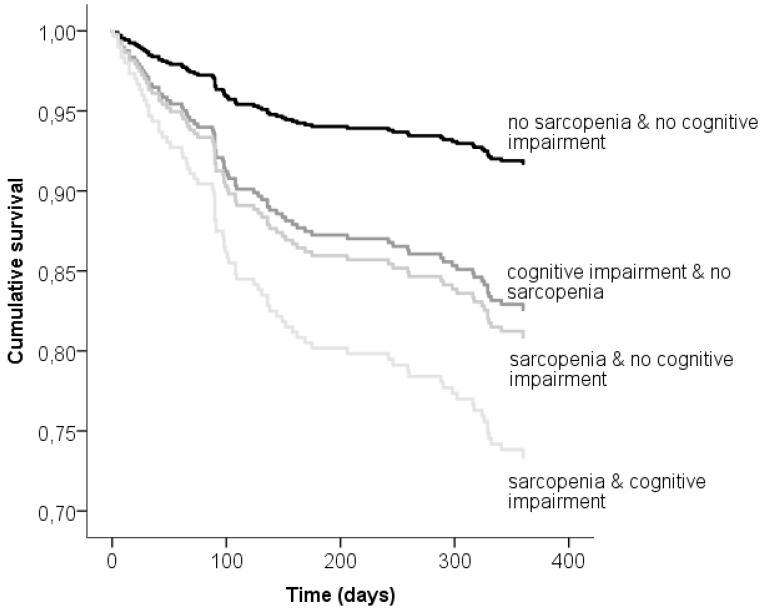
Kaplan–Meier curves showing survival according to the presence of cognitive impairment, sarcopenia or combined sarcopenia and cognitive impairment.

**Table 1 jcm-08-01693-t001:** Demographic and clinical characteristics of patients grouped according to the occurrence of sarcopenia, cognitive impairment, or sarcopenia and cognitive impairment.

	All (N = 624)	No Sarcopenia and No Cognitive Impairment (N = 231)	Cognitive Impairment and No Sarcopenia (N = 234)	Sarcopenia and No Cognitive Impairment (N = 70)	Sarcopenia and Cognitive Impairment (N = 89)	*P*
Age	80.1 ± 7.0	77.6 ± 6.6	81.2 ± 7.0	80.6 ± 7.1	83.0 ± 6.2	0.001
Gender (female)	350 (56.1)	118 (51.1)	152 (65.0)	27 (38.6)	53 (59.6)	0.001
Dependency in at least 1 BADL	168 (26.9)	25 (10.8)	93 (39.7)	10 (14.3)	40 (44.9)	0.001
Depression	201 (32.2)	63 (27.3)	87 (37.2)	24 (34.3)	27 (30.3)	0.006
History of falls	169 (27.1)	46 (19.9)	74 (31.6)	14 (20.0)	35 (39.3)	0.001
Urinary incontinence	169 (27.1)	43 (18.6)	86 (36.8)	9 (12.9)	31 (34.8)	0.001
Delirium during stay	19 (3.0)	5 (2.2)	12 (5.1)	–	2 (2.2)	0.092
BMI < 20 kg/m^2^	28 (4.5)	1 (0.4)	7 (3.0)	7 (10.0)	13 (14.6)	0.001
Length of stay	11.0 ± 6.5	10.5 ± 6.7	11.2 ± 6.1	12.4 ± 7.2	10.5 ± 6.0	0.162
Number of medications at discharge	5.3 ± 1.9	5.3 ± 1.9	5.4 ± 2.0	5.1 ± 2.1	5.0 ± 1.8	0.380
Hypertension	493 (79.0)	198 (85.7)	175 (74.8)	53 (75.7)	67 (75.3)	0.019
Coronary artery disease	192 (30.8)	66 (28.6)	74 (31.6)	25 (35.7)	27 (30.3)	0.702
Atrial fibrillation	101 (16.2)	31 (13.4)	44 (18.8)	12 (17.1)	14 (15.7)	0.467
Peripheral arterial disease	51 (8.2)	16 (6.9)	25 (10.7)	4 (5.7)	6 (6.7)	0.355
Heart failure	162 (26.0)	51 (22.1)	63 (26.9)	16 (22.9)	32 (36.0)	0.075
Cerebrovascular disease	122 (19.6)	32 (13.9)	58 (24.8)	6 (8.6)	26 (29.2)	0.001
Parkinson	35 (5.6)	8 (3.5)	16 (6.9)	4 (5.7)	7 (7.9)	0.316
Dementia	82 (13.1)	–	55 (23.5)	–	27 (30.3)	0.001
Diabetes	192 (30.8)	69 (29.9)	79 (33.8)	18 (25.7)	26 (29.2)	0.570
Chronic obstructive pulmonary disease	240 (38.5)	92 (39.8)	88 (37.6)	24 (34.3)	36 (40.4)	0.821
Malignancies	84 (13.5)	32 (13.9)	23 (9.8)	17 (24.3)	12 (13.5)	0.021
Chronic kidney disease	345 (55.3)	125 (54.1)	126 (53.8)	40 (57.1)	54 (60.7)	0.690
Arthritis/Ostheoporosis	286 (45.8)	112 (48.5)	108 (46.2)	31 (44.3)	35 (39.3)	0.522
Number of diagnoses	5.2 ± 2.7	5.0 ± 2.8	5.3 ± 2.7	5.1 ± 2.6	5.7 ± 2.7	0.220

Data are number of cases (percentage) or mean ± SD.

**Table 2 jcm-08-01693-t002:** Cox regression analysis of cognitive impairment only, sarcopenia only, or sarcopenia and cognitive impairment to 1-year mortality.

		Crude	Age- and Gender-Adjusted	Fully-Adjusted *
	Mortality, n (%)	HR (95% CI)	HR (95% CI)	HR (95% CI)
*All patients (N = 624)*				
No sarcopenia and no cognitive impairment	17 (7.4)	1.0	1.0	1.0
Cognitive impairment and no sarcopenia	31 (13.2)	2.21 (1.22–4.01)	2.15 (1.18–3.91)	1.48 (0.77–2.77)
Sarcopenia and no cognitive impairment	11 (15.7)	2.46 (1.20–5.25)	2.02 (0.98–4.14)	1.79 (0.87–3.51)
Sarcopenia and cognitive impairment	20 (22.5)	3.60 (1.88–6.85)	3.09 (1.60–5.98)	2.12 (1.05–4.13)
*Excluded patients with recorded diagnosis of dementia (N = 542)*				
No sarcopenia and no cognitive impairment	17 (7.6)	1.0	1.0	1.0
Cognitive impairment and no sarcopenia	23 (12.8)	2.10 (1.12–3.92)	2.01 (1.06–3.80)	1.50 (0.81–2.86)
Sarcopenia and no cognitive impairment	11 (15.9)	2.42 (1.14–5.18)	1.95 (0.96–4.21)	1.77 (0.80–3.70)
Sarcopenia and cognitive impairment	14 (22.6)	3.30 (1.62–6.69)	2.80 (1.35–5.74)	2.13 (1.06–4.17)
*Excluded patients with BMI <20 kg/m^2^ (N = 608)*				
No sarcopenia and no cognitive impairment	17 (7.4)	1.0	1.0	1.0
Cognitive impairment and no sarcopenia	28 (12.3)	2.05 (1.12–3.75)	1.88 (0.97–3.64)	1.44 (0.76–2.69)
Sarcopenia and no cognitive impairment	9 (14.3)	2.10 (1.03–4.70)	1.71 (0.71–3.65)	1.69 (0.72–3.44)
Sarcopenia and cognitive impairment	16 (21.1)	3.27 (1.65–6.47)	2.83 (1.23–5.17)	2.18 (1.03–3.91)
*Excluded patients with number of diagnoses >5 (N = 373)*				
No sarcopenia and no cognitive impairment	10 (6.5)	1.0	1.0	1.0
Cognitive impairment and no sarcopenia	10 (7.3)	1.34 (0.56–3.22)	1.30 (0.53–3.18)	1.03 (0.47–2.13)
Sarcopenia and no cognitive impairment	6 (14.6)	2.34 (0.95–6.43)	1.99 (0.67–5.31)	1.99 (0.84–5.66)
Sarcopenia and cognitive impairment	10 (23.8)	3.82 (1.59–9.19)	3.47 (1.38–8.69)	2.63 (1.09–6.32)
*Excluded patients with dependency in more than 4 BADL (N = 572)*				
No sarcopenia and no cognitive impairment	16 (7.0)	1.0	1.0	1.0
Cognitive impairment and no sarcopenia	24 (11.9)	2.09 (1.11–3.93)	2.08 (1.05–3.94)	1.99 (0.99–3.34)
Sarcopenia and no cognitive impairment	11 (16.2)	2.96 (1.24–5.74)	3.01 (1.28–5.56)	2.14 (1.01–4.65)
Sarcopenia and cognitive impairment	14 (19.2)	3.08 (1.50–6.31)	2.71 (1.14–4.51)	2.88 (1.10–5.73)

*Adjusted for age, gender, dependency in at least 1 BADL, depression, BMI < 20 kg/m^2^, history of falls, urinary incontinence, delirium, number of diagnoses, number of medications, and length of stay.

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
