# Peer review of "Prognosis and Interplay of Cognitive Impairment and Sarcopenia in Older Adults Discharged from Acute Care Hospitals"

_jcm, 2019, doi:10.3390/jcm8101693_

Round 1

Reviewer 1 Report

Thank you for your edits to the manuscript. 

Author Response

We want to thank the reviewer for her/his nice comments. The paper has extensively been revised by a native English speaker. We hope that the manuscript is now suitable for publication. 

Reviewer 2 Report

The manuscript could still be improved in terms of its prose but I do believe that the Reviewers have addressed the comments I had raised.

Author Response

We want to thank the reviewer for her/his nice comments. The paper has extensively been revised by a native English speaker. We hope that the manuscript is now suitable for publication. 

This manuscript is a resubmission of an earlier submission. The following is a list of the peer review reports and author responses from that submission.

Round 1

Reviewer 1 Report

Thank you for the opportunity to review this important paper.  I have only two major comments:

I would recommend to re-determine the prevalence of sarcopenia using the Revised European Consensus on definition and diagnosis (Cruz-Jentoft AJ, et al. Sarcopenia: revised European consensus on definition and diagnosis. Age Ageing. 2019 Jan 1;48(1):16-31.) Please describe how cognitive impairment was assessed - was it part of routine care competed as part of the admission, who completed the assessment?

Reviewer 2 Report

In this study the authors examined the predictive ability of combined sarcopenia and cognitive disability for all-cause mortality in a sample of 624 patients discharged from acute care wards in Italy. The findings suggest that the combination of the two increase the risk for all-cause mortality over a 1-year follow-up period compared to participants without sarcopenia or cognitive impairment, even after adjustment for relevant confounders. I have some major concerns regarding the way the interaction term was entered in the survival models and its interpretability, and some minor comments pertaining to the introduction of the manuscript and the implications of the results as described by the authors in terms of how they could be used to inform future interventions targeting people with cognitive impairment and/or sarcopenia.

In more detail:

In the Introduction, the sentence “mitochondrial dysfunction…to its development” is not very informative as it stands. The authors should expand the section on the risk factors and underlying mechanisms leading to sarcopenia over and above the normal reductions in muscle mass that can be attributed to ageing. They should also elaborate on the mechanisms linking cognitive impairment with all-cause mortality and the relationships between cognition and sarcopenia. Overall, the introduction is too short to adequately introduce the topic. Moreover, the sentence “However, the prognostic … investigated until now” does not follow conceptually the preceding sentences. The introduction should lead to the identification of the gaps in the literature, and the hypotheses formulated, a lot smoother. The authors might find a recent article by Cruz-Jentoft et al. (2018) introducing revised criteria for sarcopenia (EWGSOP2) particularly useful.

Cruz-Jentoft, A. J., Bahat, G., Bauer, J., Boirie, Y., Bruyère, O., Cederholm, T., ... & Schneider, S. M. (2018). Sarcopenia: revised European consensus on definition and diagnosis. Age and ageing, 48(1), 16-31.

In the results: “sensitivity analyses carried out after excluding patients with recorded diagnosis of dementia, underweight, high comorbidity burden and severe disability”. How many were these patients? I suspect that this analysis could be severely under-powered (albeit very informative).

This is my main concern. In the Methods, the authors state that “The interaction term sarcopenia*cognitive impairment was also included in the model.” However, this is not to be found in Table 2, and the authors do not comment on it in the Results. In order to be able to conclude if the interaction between cognitive impairment and sarcopenia is a significant predictor of all-cause mortality over and above cognitive disability or sarcopenia independently, the interaction term as well as the individual components should be predictors in the Cox model.

On a similar note, by using the categorisation presented in Table 2 regarding the presence of cognitive disability, sarcopenia, or both, the reference group is a group of patients without any of the two. I would recommend that the authors at least attempt to re-run the models after changing the reference category to ‘Cognitive impairment and no sarcopenia’ and also to ‘Sarcopenia and no cognitive impairment’ in order to be able to contrast the combined group against these two. In that way we could compare and contrast the relative predictive ability of cognitive impairment AND sarcopenia, against having one OR the other.

Also, it should be made very clear in the abstract that the combination of sarcopenia and cognitive ability was tested against participants with intact cognitive ability and without sarcopenia.

In the discussion: “At the same time… reported [28].” In line with my second comment, this information should be part of the introduction. In addition, the mechanisms described in the second paragraph of the discussion could have been part of the introduction.

Consider revising the paragraph “Whatever is the … future investigations”. It is difficult to follow. Also, what could potential interventions targeting both, cognitive impairment and sarcopenia, entail?

Consider revising the sentence “Thus, our findings… by specific interventions”. First, it is difficult to follow. Most importantly, I do not agree that the findings presented can be used to describe “a condition of particular vulnerability”. I think that the statement is pushing the implications of the study too far as it stands.

“patients’ acute conditions related to hospitalization may have contributed to an overestimation of the diagnosis of sarcopenia and cognitive impairment in the present sample”. Why would that be?

Minor syntax and grammar errors are scattered throughout the manuscript. The study would benefit from an extensive revision to amend these. Just to name a few: a) in the abstract “prognostic interplay” and “prognostic interaction” should read “the predictive ability of combined sarcopenia and cognitive impairment”; b) in the introduction revise “sarcopenia was found associated”; etc.